# The Increase in FGF23 Induced by Calcium Is Partially Dependent on Vitamin D Signaling

**DOI:** 10.3390/nu14132576

**Published:** 2022-06-22

**Authors:** Sandra Rayego-Mateos, Nuria Doladé, Alicia García-Carrasco, Juan Miguel Diaz-Tocados, Merce Ibarz, Jose Manuel Valdivielso

**Affiliations:** 1Red de Investigación Renal (REDinREN), 28029 Madrid, Spain; srayego@quironsalud.es (S.R.-M.); nuriadolade@gmail.com (N.D.); agarcia@irblleida.cat (A.G.-C.); 2Vascular and Renal Translational Research Group, Institut de Recerca Biomèdica de Lleida IRBLleida, 25198 Lleida, Spain; jmdiaz@irblleida.cat; 3Indicators and Specifications of the Quality in the Clinical Laboratory Group, IRBLleida, 25198 Lleida, Spain; mibarz.lleida.ics@gencat.cat

**Keywords:** FGF23, bone, kidney, calcium, BMD-CKD

## Abstract

Background: Increased FGF23 levels are an early pathological feature in chronic kidney disease (CKD), causing increased cardiovascular risk. The regulation of FGF23 expression is complex and not completely understood. Thus, Ca^2+^ has been shown to induce an increase in FGF23 expression, but whether that increase is mediated by simultaneous changes in parathyroid hormone (PTH) and/or vitamin D is not fully known. Methods: Osteoblast-like cells (OLCs) from vitamin D receptor (VDR)^+/+^ and VDR^−/−^ mice were incubated with Ca^2+^ for 18 h. Experimental hypercalcemia was induced by calcium gluconate injection in thyro-parathyroidectomized (T-PTX) VDR ^+/+^ and VDR^−/−^ mice with constant PTH infusion. Results: Inorganic Ca^2+^ induced an increase in FGF23 gene and protein expression in osteoblast-like cells (OLCs), but the increase was blunted in cells lacking VDR. In T-PTX VDR ^+/+^ and VDR^−/−^ mice with constant PTH levels, hypercalcemia induced an increase in FGF23 levels, but to a lower extent in animals lacking VDR. Similar results were observed in FGF23 expression in bone. Renal and bone 1α-hydroxylase expression was also modulated. Conclusions: Our study demonstrates that Ca^2+^ can increase FGF23 levels independently of vitamin D and PTH, but part of the physiological increase in FGF23 induced by Ca^2+^ is mediated by vitamin D signaling.

## 1. Introduction

Mineral metabolism regulation is a very complex process that aims to control calcium (Ca^2+^) and phosphate (P) levels in serum through the interaction of three hormones and three body systems. Thus, parathyroid hormone (PTH), vitamin D and fibroblast growth factor 23 (FGF23) interact with bones, the kidneys and the intestine to maintain adequate levels of Ca^2+^ and P in plasma. Ca^2+^ and P metabolism are closely related and highly dependent on one another. Both ions are regulated by 1,25-dihydroxyvitamin-D (1,25(OH)_2_D) and PTH [1]. Thus, 1,25(OH)_2_D increases the absorption of Ca^2+^ and P in the intestine, whereas PTH increases Ca^2+^ and P extraction from bone and Ca^2+^ reabsorption in the kidney. Finally, FGF23, secreted by osteoblasts and osteocytes, has been shown to decrease P reabsorption in the kidney [2,3]. 

Although P is a central component of nucleic acids, cell membranes and energy metabolism [4], excessive P levels are associated with increased mortality and vascular calcification [5,6,7]. Thus, the main function of FGF23 seems to be the protection of the organism against the deleterious effects of hyperphosphatemia by inhibiting the sodium-dependent phosphate cotransporters in the kidney, decreasing intestinal absorption through the suppression of renal synthesis of 1,25(OH)_2_D and decreasing bone reabsorption by inhibiting PTH synthesis and release [2,3,8,9,10]. Those renal effects are achieved through the interaction with the fibroblast growth factor receptor (FGFR) and its coreceptor α-Klotho [11,12].

In normal homeostasis, the three hormones and the two ions are kept in a constant equilibrium, which maintains their serum levels within a tight range. However, when kidney function starts to fail, the balance is lost, leading to a complication called chronic kidney disease–mineral bone disorder (CKD-MBD). CKD-MBD is a clinical syndrome encompassing mineral, bone and calcific cardiovascular abnormalities associated with increased mortality and fractures in CKD patients. In the course of CKD-MBD, one of the first biochemical alterations found is an increase in FGF23 levels in serum [13]. Elevated FGF23 has been identified as an independent marker for cardiovascular risk in different populations. Therefore, understanding the regulation of FGF23 expression is of paramount importance in order to decrease morbimortality in CKD.

The underlying mechanisms of the rise of FGF23 in renal insufficiency remain partly unclear. Various endogenous and external factors contribute to FGF23 regulation, such as the phosphorus load and high 1,25(OH)_2_D levels, which seem to be the main stimulators of FGF23 synthesis [14,15]. Other factors, including calcium, parathyroid hormone, inflammation and iron, are also involved [16]. The regulation of FGF23 synthesis by calcium is a process not fully understood. Previous studies have shown a PTH-independent calcium effect on FGF23 protein levels in parathyroidectomized or PTH/Gcm2 KO animals with a high-calcium diet or calcium gluconate injection for one week [1,17,18]. However, no change in FGF23 protein levels was found in acute calcium changes in parathyroidectomized rats [19]. Furthermore, and despite of its activity regulating FGF23, data point to an irrelevant role of vitamin D in Ca^2+^-induced FGF23 regulation [18,20]. However, the intricate relationship between all the players in the regulation of mineral metabolism precludes a clear understanding of how Ca^2+^ regulates FGF23. Therefore, the present study was designed to determine whether Ca^2+^ can regulate FG23 levels independently of simultaneous changes in PTH and vitamin D.

## 2. Materials and Methods

### 2.1. Study in Animals

Mouse experiments were performed in adult B6CBA mice or VDR-deficient (VDR^−/−^) mice on a B6CBA genetic background. All procedures on animals were performed according to the recommendations of the European Research Council for the Care and Use of Laboratory Animals, and the protocol was approved by the local Animal Ethics Committee of the University of Lleida (CEEA 07-02/14).

#### 2.1.1. Generation of VDR^−/−^ Mice

VDR-deficient (VDR^−/−^) mice (a kind gift from Dr. Kato, University of Tokyo, Japan) were backcrossed more than 8 times to C57BL/6J mice and have since been maintained in our colony for more than 7 years. To confirm the genotype, the animals were genotyped for VDR by PCR analysis of tail DNA. Mice obtained from this crossbreeding and used in our experiments had a mixed B6CBA background. All animals were weaned at 21 days. After weaning, VDR^+/+^ mice were maintained on a regular mouse chow (Harlan Teklad, WI, USA), while VDR^−/−^ mice were fed a high-calcium, high-phosphate diet (rescue diet, TD.96348, 20% Lactose, 2% Ca, 1.25% P; Harlan Teklad) to prevent hypocalcemia. 

#### 2.1.2. Thyro-Parathyroidectomy

Parathyroid gland removal was performed by electro-cauterization in VDR^+/+^ and VDR^−/−^ C57BL/6J mice (*n* = 10 per group). As a side effect, most of the thyroid gland was also removed. The surgery was performed under general anesthesia (Isoflurane) in male or female mice (8–10 weeks old). After the surgery mice received pain medication (buprenorphine, 0.05 mg/kg, sc). Animals also received a replacement T4 treatment every day (L-thyroxine, 40 ng/g, sc, Sigma-Aldrich, St. Louis, MO, USA) until sacrifice, as most of the thyroid gland was also cauterized to assess complete parathyroid removal. In order to confirm the correct ablation of the parathyroid gland, we measured the Ca^2+^ and PTH levels in total blood and serum of mice subjected to gland ablation 10 days after the surgery. 

#### 2.1.3. PTH (1–34) Infusion

After parathyroid gland ablation, mice were implanted subcutaneously with osmotic minipumps (ALZET 1002; 0.25 μL/h) to maintain constant (1–34) PTH levels (0.20 ug/100 g/h, GeneScript, NJ, USA). 

#### 2.1.4. Experimental Hypercalcemia 

Three days after the osmotic minipump implantation, animals were submitted to hypercalcemic treatment with subcutaneous injection of calcium gluconate monohydrate (250 mg/kg) every 2 h for 8 h. After that, mice were euthanized with 5 mg/kg xylazine (Rompun, Bayer; Leverkusen, Germany) and 35 mg/kg ketamine (Ketolar, Pfizer, NY, USA), and tissues were perfused with cold PBS through a puncture in the left ventricle. Then, all tissue portions were fixed in buffered formalin for immunohistochemistry studies or immediately frozen in liquid nitrogen for gene and protein studies. Blood samples were collected by cardiac puncture with heparinized syringes to analyze urea and creatinine plasma levels, and urine was collected to evaluate proteinuria.

### 2.2. In Vitro Studies 

To obtain bone-marrow-derived mesenchymal stem cells (MSCs), 8–10 mice per group were euthanized with 5 mg/kg xylazine (Bayer, AG, USA) and 35 mg/kg ketamine (Pfizer, NY, USA). For MSC extraction, both tibiae and femurs of each mouse were cut at the epiphyses and subsequently perfused with alpha minimal essential medium (αMEM; Sigma-Aldrich, MO, USA) containing FBS (15%; Sigma Aldrich), ultraglutamine (1%; Lonza Inc., Walkersville, MD, USA), penicillin (100 U/mL) and streptomycin (100 µg/mL) under sterile conditions. Once the cells were obtained, they were filtered through a 70 µm cell strainer (Corning, NY, USA). Bone marrow cells were centrifuged and washed two times with αMEM before being cultivated in 25 cm^2^ flasks (Corning) with αMEM containing FBS (15%), ultraglutamine (1%), penicillin (100 U/mL), streptomycin (100 µg/mL) and basic fibroblast growth factor (bFGF; 1 ng/ mL; PeproTech, London, UK) in a humidified atmosphere with 5% CO_2_ at 37 °C. Fresh α-MEM with FBS (15%), ultraglutamine (1%), penicillin (100 U/mL), streptomycin (100 µg/mL) and bFGF (1 ng/mL) was added after 24 h and changed every 3 days. Once 85–90% confluence was reached, cells were collected using Trypsin-EDTA (Lonza) and seeded in 6-well plates (Corning) at 13,000 cells/cm^2^. 

When cells reached 80% confluence, they were differentiated into osteoblast-like cells (OLCs). To do so, cells were cultured in αMEM with FBS (10%), ultraglutamine (1%), penicillin (100 U/mL), streptomycin (100 µg/mL) and osteogenic stimuli based on dexamethasone (1 µM; Sigma-Aldrich), β-glycerol phosphate (10 mM; Sigma-Aldrich) and ascorbic acid (0.2 mM; BAYER, Barcelona, Spain) for 21 days. We developed two different groups of cells for the osteogenic differentiation study: (1) Control MSCs: MSCs without differentiation medium for 21 days; (2) OLCs that were MSCs with differentiation medium for 21 days. To confirm a correct differentiation process, different osteogenic genes were evaluated. 

The OLCs were stimulated with CaCl_2_ (6 and 8 mM; Sigma-Aldrich) or a 10 mM inorganic phosphorous (P_i_) mixture 1:2 (NaH_2_PO_4_ + Na_2_HPO_4_, respectively; Sigma-Aldrich), with a final concentration of 6 mM and 8 mM for calcium (CaCl_2_) and 10 mM for P_i_, for 18 h.

### 2.3. Biochemical Analyses

After total blood extraction through heparinized syringes, Ca^2+^ levels were immediately analyzed in total blood using the GEM^®^ Premier 4000 gasometer analyzer (Werfen, Barcelona, Spain). After that, plasma was isolated by centrifugation and stored at −80 °C. Plasma and urine Ca^2+^ and P levels were measured by Epoch spectrophotometer (BioTek Instruments, Winooski, VT, USA) using a phosphorus measurement kit (Biosystems, Barcelona, Spain) and following the manufacturer’s protocol. ELISA was used to evaluate levels of PTH (1–84) and i-FGF23 in plasma (Immutopics, San Diego, CA, USA) following the manufacturer’s protocol. 

### 2.4. Bone and Renal Histology, Immunohistochemistry

At sacrifice, both femurs and tibiae were placed in 70% ethanol and stored at room temperature until processing. Femurs were embedded in 75% methyl methacrylate, 25% dibutyl phthalate, and 2.5% *w*/*v* benzoyl peroxide with a previous dehydration in alcohol/xylene. Bone slices (5 mm) were deacrylated in a 1:1 mixture of xylene and chloroform for 30 min and rehydrated with graded ethanol concentrations (100°, 95°, 70°) and distilled water. Afterwards, samples were decalcified with 14% EDTA, pH 7.4 for 1 h and washed for 10 min with distilled water. Moreover, 3 μm thick kidney sections were used to analyze 1-α hydroxylase expression. 

Immunohistochemistry was then carried out using conventional methods with the subsequent steps: (1) Endogenous peroxidase blockade (30 min incubation in 0.3% (*v*/*v*) H_2_O_2_/PBS); (2) Non-specific binding blockade with 3% BSA/PBS for 1 h at room temperature; (3) Incubation with primary antibody anti-FGF23 (15 μg/mL; Biotechne, Minneapolis MN, USA) and 1 a hydroxylase (1:100; Cloud Clone, Katy, TX, USA) overnight at 4 °C; (4) Washing in PBS; (5) Incubation with 1:100 dilution of corresponding biotinylated secondary antibody (anti-rat and anti-rabbit, Vector Labs, Newark, CA, USA) for 1 h at room temperature; (6) Incubation with avidin–biotin–peroxidase complex (Vector Labs) previously prepared and stored in the dark for 30 min; (7) Addition of 3,3′-diaminobenzidine as chromogen (Vector Laboratories). Sections were counterstained with hematoxylin. Negative controls were obtained with PBS incubation instead of primary antibody following by the same secondary antibody incubation (not shown). Images were taken using an Olympus BX50 microscope with an Olympus automatic camera system. Specificity was checked by omission of primary antibodies (not shown). Quantification was made by determining the positive staining in 5–10 randomly chosen fields (×200 magnification) relative to the total area using Image-Pro Plus software (ciudad), or by manually counting positive staining.

### 2.5. Immunofluorescence Staining of FGF23

Paraffin-embedded kidney sections (3 μm) were submitted for antigen retrieval. After the slides were blocked with 10% BSA and 10% FBS for 1 h, they were incubated with FGF23 primary antibody (1/200) for 1 h, followed by a AlexaFluorTM 594 conjugated secondary antibody (1/200; Invitrogen, Waltham, MA, USA) for 1 h. The absence of primary antibody was used as negative control. Samples were mounted in moviol and examined using a Leica DM-IRB confocal microscope.

### 2.6. Red Alizarin Staining and Absorbance Measurement 

Matrix mineralization was detected using this protocol. The cells were fixed with paraformaldehyde (2%) and sucrose (1%) for 15 min. After that, the cells were stained with alizarin red S pH 4.1 (40 mM, Sigma-Aldrich) for 20 min and subsequently washed 4 times for 5 min with water at pH 7. After the cell plates were dried at room temperature, 800 μL of Acetic Acid 10% (*v*/*v*) was added to each well, and the plates were incubated at RT for 30 min under agitation. The cells were scraped and transferred to an Eppendorf and heated at 85° for 10 min. At the end, the tubes were centrifuged at 20,000 *g* for 15 min, and 200 μL of ammonium hydroxide 10% (*v*/*v*) was added to neutralize the acid. Finally, the absorbance was read at 405 nm using an Epoch spectrophotometer (BioTek Instruments).

### 2.7. Protein Extraction and Western Blot Analysis 

Total protein was isolated from cells or mouse organs through lysis buffer containing 50 mmol/L Tris-hydrochloride, 150 mol/L sodium chloride, 2 mmol/L EDTA, 2 mmol/L EGTA, 0.2% Triton X-100, 0.3% IGEPAL, 10 μL/mL proteinase inhibitor cocktail, 0.2 mmol/L PMSF and 0.2 mmol/L orthovanadate. Protein concentrations were determined using a DC protein assay kit (Bio-Rad, CA, USA). For Western blot, 10–20 μg of protein extracts were used and separated in 12% SDS-PAGE gels under reducing conditions. Samples were subsequently transferred to PDVF membrane (pore size 0.45 µm, Immobilon-P, Millipore, Massachusetts, MA, USA) and blocked with 5% defatted milk diluted in Tris-buffered saline with 0.01% Tween 20 for 1 h at room temperature. Afterwards, membranes were washed and incubated overnight at 4 °C with specific FGF23 (1:1000, Biotechne, Minneapolis, MN, USA) and GAPDH (1:10,000, Abcam, Cambridge, UK) antibodies. Horseradish peroxidase-conjugated secondary antibodies (anti-rat and anti-mouse, Jackson Immuno Research Labs, West Grove, PA, USA) were used at 1/10,000 for 1 h at room temperature, and membranes were exposed using the chemiluminescent kits EZ ECL (Biological Industries, Beit Haemek, Israel) or ECL Advanced (Amersham Biosciences, Amersham, UK). Images were digitally acquired by ChemiDoc™ MP Imaging System (Bio-Rad, Hercules, CA, USA) and analyzed with Quantity one 1-D analysis software (Bio-Rad). Data are indicated as n-fold increase over control mice, as mean ± SEM of 8–10 animals per group.

### 2.8. Gene Expression Studies 

Total RNA was isolated from cells or mouse organs using TRIzol reagent (Sigma), and cDNA was synthesized with the First Strand cDNA Synthesis Kit (AMV) (Roche, Basilea, Switzerland) following the manufacturer’s instructions. Singleplex real-time PCR was performed with the CFX Real-Time PCR detection system (Bio-Rad) using the following specific TaqMan probes for murine samples (ThermoFisher, Waltham, MA, USA): FGF23 (Mm_00445621_m1), 1α-hydroxylase (Mm01165918_g1), RUNX2 (Mm00501584_m1), Osteopontin (Mm00436767_m1) and Osterix (Mm04933803_m1). Data were normalized to tbp (Mm00446971_m1, ThermoFisher) expression levels. The rt-PCR was performed using TaqMan Universal PCR Master Mix, No AmpErase UNG (ThermoFisher) with forty cycles at 95 °C for 15 s and 60 °C for 1 min. The mRNA copy numbers were calculated by standard formulae (ΔΔCt method). Results were normalized against TBP expression levels and are expressed as relative changes with respect to unstimulated cells or control mice.

### 2.9. Statistical Analyses

Results were analyzed with GraphPad Prism (GraphPad Software, San Diego, CA, USA). All data are expressed as mean ± SEM. Differences between treated groups and controls were evaluated using the Student’s *t*-test. Differences were considered significant when *p* < 0.05.

## 3. Results

### 3.1. Lack of VDR in Osteoblast-like Cells (OLCs) Blunted the Stimulation of FGF23 Induced by Inorganic Calcium

To analyze the modulation of FGF23, we needed to establish the correct conditions for a primary cell culture of bone-marrow-derived mesenchymal stem cells (MSCs) from VDR^+/+^ and VDR^−/−^ mice. For that, we differentiated those cells into osteoblast-like cells (OLCs) using an osteogenic differentiation medium. We evaluated the success of the culture by checking the formation of a mineral matrix using red alizarin staining (Figure 1B) and modifications in cell morphology (Figure 1A), and by measuring the increase in gene expression of osteogenic markers (RUNX2, OSTEOPONTIN, OSTERIX, OSTEOCALCIN or FGF23) in the differentiated cells (Figure 1C). Osteoblast-like cells showed clear differences in cell morphology compared to MSCs, mainly losing fibroblastic and spindle-shaped morphological features and acquiring a cuboidal or polygonal shape (Figure 1A). Moreover, in the OLCs, the formation of calcium nodules (mineralization) assessed by red alizarin staining (Figure 1B,D) was observed, as well as increased gene levels of osteogenic markers, such as RUNX2, OSTEOPONTIN, OSTERIX, OSTEOCALCIN and FGF23, in comparison with non-differentiated MSCs ( Figure 1C). In addition, we also observed an increase in protein levels of FGF23 in differentiated cells from VDR^+/+^ mice, which was slightly lower in cells obtained from VDR^−/−^ mice, assessed by immunofluorescence and Western blot, respectively (Figure 1D,F).

To analyze the capacity of inorganic calcium to induce FGF23 gene and protein expression in bone cells, we checked the effect of calcium chloride (CaCl_2_) stimulation in OLCs coming from the differentiated bone marrow MSCs of VDR^+/+^ and VDR^−/−^ mice after 18 h (Figure 2). We also stimulated the OLCs with inorganic phosphorus (P_i_: NaH_2_PO_4_ + Na_2_HPO_4_) as a control to test the positive regulation of FGF23 by phosphorus. CaCl_2_ and Pi stimulation of the OLCs modified the phenotype of the VDR^+/+^ and VDR^−/−^ cells and reduced cell viability, as observed using bright field (Figure 2A,D). Through fluorescence microscopy, it was observed that the treatment with CaCl_2_ and P_i_ increased the protein expression of FGF-23 in VDR^+/+^ and VDR^−/−^ OLCs in comparison with control cells (OLCs without stimulation) (Figure 2A). 

In addition, in VDR^+/+^ OLCs, CaCl_2_ stimulation significantly increased FGF23 gene expression and protein levels compared to unstimulated OLCs after 18 h, and the FGF23 levels increased more than the levels induced by P_i_ (Figure 2B,C). A similar increase in FGF23 levels was observed in VDR^−/−^ OLCs after CaCl2 stimulation (Figure 2E,F). Importantly, the effect of CaCl_2_ on FGF23 protein expression was blunted (around 50%) in VDR^−/−^ OLCs compared to VDR ^+/+^ OLCs after 18 h (Figure 2C,F). These results in vitro demonstrated that some of the effects of Ca^2+^ increasing the synthesis of FGF23 in bone are mediated through VDR.

### 3.2. The Stimulation with Inorganic Calcium in VDR^+/+^ and VDR^−/−^ OLCs from Mice Increases the Expression of 1α-Hydroxylase

The enzyme responsible for the production of 1,25(OH)_2_D is 25-Hydroxyvitamin D 1α-hydroxylase. Although 1α-hydroxylase is expressed predominantly in the kidney, extra-renal production has also been described in other tissues such as bone. We wanted to analyze whether Ca^2+^ and P_i_ levels modulate 1α-hydroxylase expression in OLCs from VDR^+/+^ and VDR^−/−^ mice. VDR^+/+^ and VDR^−/−^ OLCs stimulated with CaCl_2_ and Pi for 18 h showed an increase in 1α-hydroxylase gene expression levels (Figure 3A,C respectively). Again, it seemed that the effect of the highest concentration of CaCl_2_ was higher than the effect of Pi in VDR^+/+^ cells, whereas that tendency was lost in VDR^−/−^ cells. 

The proteins levels of 1α-hydroxylase were diminished in VDR^+/+^ and VDR^−/−^ OLCs stimulated by CaCl_2_ and P_i_ (Figure 3B,D), suggesting a possible post-transcriptional modification.

### 3.3. Experimental Hypercalcemia Induced in VDR^+/+^ and VDR^−/−^ Mice, Increases in FGF23 Is Partially VDR-Dependent

To evaluate in vivo the role of Ca^2+^ in the synthesis of FGF23, we established mouse models of calcium overload induced by calcium gluconate monohydrate (C_12_H_22_CaO_14_·H_2_O) injection. In serum samples from VDR^+/+^ injected mice, modifications in calcium levels were not observed (Figure 4A). However, a significant increase in urinary calcium excretion demonstrated an increase in the Ca^2+^ load (Figure 4B). In order to avoid fluctuations in PTH levels during the study, we eliminated the parathyroid gland (and most of the thyroid gland as a side effect). A correct parathyroidectomy (T-PTX) was confirmed by a decrease in ionic Ca^2+^ in plasma, together with reduced PTH (Figure 4C,D).

Once T-PTX was established, VDR^+/+^ and VDR^−/−^ mice were implanted subcutaneously with an osmotic minipump filled with human PTH (1–34) to maintain constant PTH levels during the study. After 8 h of calcium gluconate monohydrate injection, T-PTX VDR^+/+^ and VDR^−/−^ mice showed an increase in plasma ionic Ca^2+^ levels compared to control mice (T-PTX VDR^+^/^+^ and VDR^−/−^ mice not injected) (Figure 5A), without changes in the serum P (Figure 5B). Intact FGF23 levels in the plasma of T-PTX VDR^+/+^ and VDR^−/−^ mice were increased, but to a lower extent in animals lacking VDR (Figure 5C). Similar results were observed for the FGF23 expression in bone (Figure 5D,G). The increase in FGF23 strongly correlated with serum Ca^2+^ levels (Figure 5E), but it did not with P levels (Figure 5F).

### 3.4. Experimental Hypercalcemia Induced in VDR^+/+^ and VDR^−/−^ Mice Modulates Renal and Bone 1α-Hydroxylase Gene and Protein Levels

To evaluate the role of Ca^2+^ in the modulation of 1α-hydroxylase independently of PTH levels and VDR, we used the hypercalcemic model induced by calcium gluconate monohydrate injection described above. In VDR^+^/^+^ mice, calcium overload decreased the 1α-hydroxylase gene and protein levels in bone (Figure 6A,B) and in renal tissue (Figure 6C,D). In addition, in VDR^−/−^ mice, hypercalcemia showed a significant decrease in the 1α-hydroxylase protein levels in bone, but not in its gene expression (Figure 6A,B), and only a tendency to decrease it in the kidney (Figure 6C).

## 4. Discussion

In the present study, we aimed to understand whether the effect of Ca^2+^ regulating FGF23 levels was dependent on changes of the other two main hormones involved in its regulation, namely, PTH and vitamin D. The results show that, in a model of hypercalcemia with constant levels of PTH, FGF23 regulation by calcium is partly dependent on changes in vitamin D signaling.

The regulation of FGF23 expression in bone is still not completely understood. Since its discovery in 2001 as a phosphaturic hormone [21] and results showing its accumulation in plasma from early stages of CKD [22], the investigation into the regulation and possible health implications of elevated levels of FGF23 has increased exponentially. Thus, early studies from Dr. Wolf’s laboratory showed that, although increased FGF23 levels may prevent hyperphosphatemia, they caused an over-suppression of active vitamin D synthesis, which could have detrimental consequences [23]. Indeed, subsequent studies from the same group demonstrated that elevated FGF23 levels could be behind the increased mortality observed in CKD patients [24,25]. Furthermore, increases in FGF23 were targeted as the earliest alterations in CKD-MBD [13], so understanding the mechanisms involved in its upregulation can lead to early intervention strategies in CKD.

The first hypothesis was that, as FGF23 is a phosphaturic hormone, increased P levels would likely stimulate its synthesis. However, it has been demonstrated that an increase in P load in the diet will increase FGF23 levels without changes in blood P [26], so additional mechanisms for its regulation are in place. Posterior data have shown that the rest of the players in mineral metabolism, namely, PTH, active vitamin D and Ca^2+^, regulate FGF23 expression and release. Thus, the FGF23 promoter region shows a response element for vitamin D, which increases its expression [27]. Furthermore, PTH also directly and indirectly (through increasing active vitamin D synthesis) increases FGF23 synthesis and release [28]. Furthermore, an effect of Ca^2+^ on FGF23 synthesis has been suggested [18], but whether that effect is partly mediated by Ca^2+^-induced changes in vitamin D or PTH is not fully known.

In our study, we used primary bone multipotent stem cells differentiated into osteoblasts-like cells. Thus, the differentiated cells showed the genotypic and phenotypic characteristics of osteoblasts, expressing osteoblastic genes and inducing calcification of the extracellular matrix. Furthermore, the differentiated cells showed an increased expression of FGF23. However, the increase was lower in cells obtained from animals lacking VDR (VDR^−/−^ mice), pointing to a pivotal role of vitamin D in FGF23 expression in bone, as has been previously suggested [20]. When we incubated these cells with Ca^2+^ (CaCl_2_) or P_i_ (NaH_2_PO_4_ + Na_2_HPO_4_), the results indicated that P_i_ showed a tendency to increase FGF23 expression, as previously described [2], but CaCl_2_ further increased FGF23 expression levels. Our results agree with those of David et al., which showed that, in a pre-osteoblastic cell line, the addition of Ca^2+^ to the culture media increased the expression of FGF23 [18]. Our results further showed that, in cells without vitamin D signaling, the effect of Ca^2+^ on FGF23 expression was partially abolished, so the direct effect of Ca^2+^ on FGF23 expression in osteoblasts is mediated, at least in part, by vitamin D signaling. Thus, in VDR^+^/^+^ cells, incubation with high Ca^2+^ led to a decrease in 1 α-hydroxylase protein levels (although a slight increase in mRNA was detected). This fact points to a direct effect of extracellular Ca^2+^ decreasing active vitamin D synthesis in bones, as it has been described systemically [29]. However, when vitamin D signaling was absent (VDR^−/−^ cells), the decrease in local vitamin D synthesis was blunted, and this mechanism could be involved in the lower induction of FGF23 seen in vitro in a VDR-independent way. Indeed, VDR-independent effects of active vitamin D in osteoblasts have been demonstrated on nitric oxide production, Ca^2+^ mobilization and activation of phospholipase C [30,31,32].

In our in vivo model, we found that, in animals with constant levels of PTH, hypercalcemia induced a higher increase in FGF23 levels when vitamin D signaling was intact. Thus, after injection with calcium gluconate, both VDR^+/+^ and VDR^−/−^ animals showed significant increases in Ca^2+^ in plasma, with no changes in P. However, the blood levels and the bone expression of FGF23 observed in the VDR^+/+^ animals was twice that reached in the VDR^−/−^ mice, confirming the in vitro findings that point to a paramount role of vitamin D signaling in the increase in FGF23 induced by Ca^2+^. In previous studies, the effect of Ca^2+^ on the regulation of FGF23 has been tested. However, this is the first time in which the effect was tested independently of changes in both PTH and vitamin D. In a previous study, the role of Ca^2+^ in FGF23 expression was tested independently of PTH and vitamin D separately. However, the very close relationship between these two hormones makes it very difficult to interpret the results. Thus, David et al. induced hypercalcemia in Cyp27b1^−/−^ mice, which cannot synthetize endogenous vitamin D, and in Gcm2^−/−^ mice, which cannot synthesize PTH. A high-Ca^2+^ diet also increased serum FGF23 concentrations in both mice, but to a very different extent. Furthermore, the administration of Ca^2+^ to the Cyp27b1^−/−^ mice very strongly decreased the basal PTH levels in those animals, which probably had a role in the changes in FGF23. In the same line of reasoning, the administration of a high-Ca^2+^ diet would have probably changed the levels of active vitamin D in the Gcm2^−/−^, a fact that was not tested in the paper. Indeed, the authors stated that the design of the in vivo study could not determine whether the effects of Ca^2+^ on FGF23 production were direct or indirect [18]. In another study in parathyroidectomized rats, hypocalcemia induced by a low-calcium diet and hypercalcemia induced by infusion of Ca^2+^ decreased and increased, respectively, the levels of FGF23, together with opposite changes in the circulating active vitamin D levels, probably attenuating the real effect of Ca^2+^ on FGF23. Indeed, the administration of Ca^2+^ in our model decreased the expression of 1α-hydroxylase both in the kidney and bone in vivo, probably decreasing the active vitamin D production and playing a role in the regulation of FGF23 by calcium. Thus, a possible physiological explanation of the whole system would be as follows. When high Ca^2+^ levels are detected in vitro, FGF23 levels are increased. The levels reached are dependent on several stimulatory pathways controlled by PTH, vitamin D, Ca^2+^ and P. With fixed PTH levels, if both Ca^2+^ and vitamin D levels are increased, the increase in FGF23 is higher, probably as a defense mechanism to avoid hyperphosphatemia by decreasing both P levels and vitamin D synthesis. However, if Ca^2+^ levels are high but Vitamin D signaling is low, the defense mechanism against hyperphosphatemia is not triggered to its full potential, and the FGF23 levels do not increase as much. 

## 5. Conclusions

In conclusion, our study demonstrates that Ca^2+^ can stimulate FGF23 synthesis and release independently of vitamin D and PTH changes. However, the physiological increase in FGF23 induced by Ca^2+^ is partially mediated by changes in vitamin D signaling.

## Figures and Tables

**Figure 1 nutrients-14-02576-f001:**
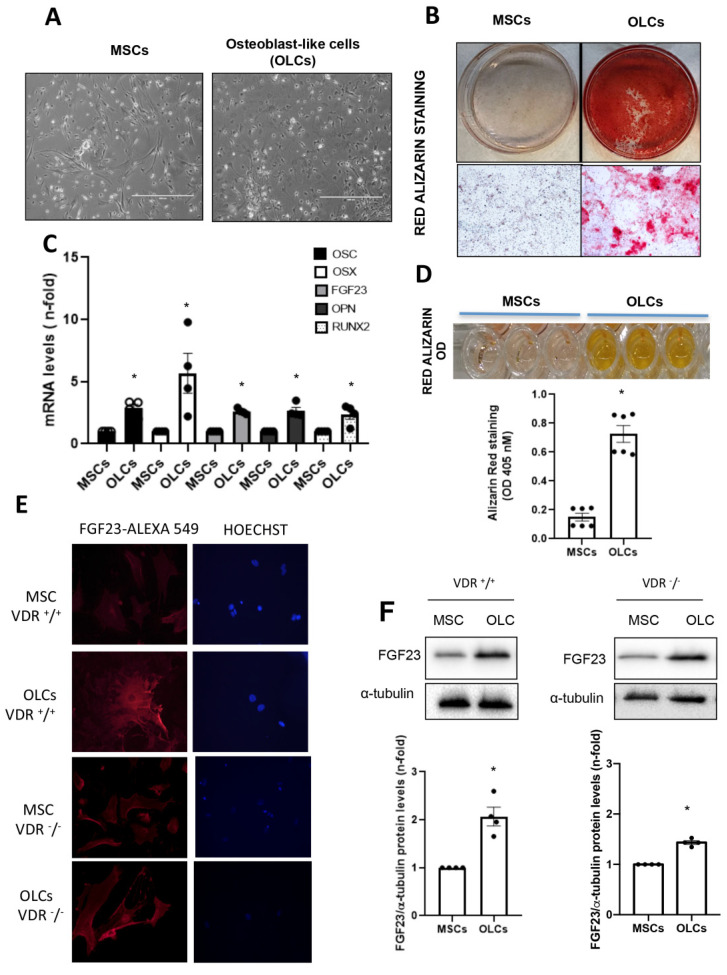
Establishment of an in vitro model to study FGF23 expression. Primary culture of mesenchymal stem cells (MSCs) from bone marrow of VDR^+/+^ and VDR^−/−^ mice and MSCs differentiated to OLCs by culture for 21 days in an osteogenic differentiation medium. (**A**) Phenotypic differences between MSCs and OLCs by bright field microscopy. (**B**) Red alizarin staining to identify the mineral matrix accumulation. (**C**) Gene expression of osteogenic and mesenchymal marker genes by RT-PCR. (**D**) Quantification of red alizarin staining by optical density. (**E**,**F**) FGF23 production by mature osteoblasts assessed by immunofluorescence (**E**) and Western blot (**F**). * *p* < 0.05 vs. OLC without stimulus.

**Figure 2 nutrients-14-02576-f002:**
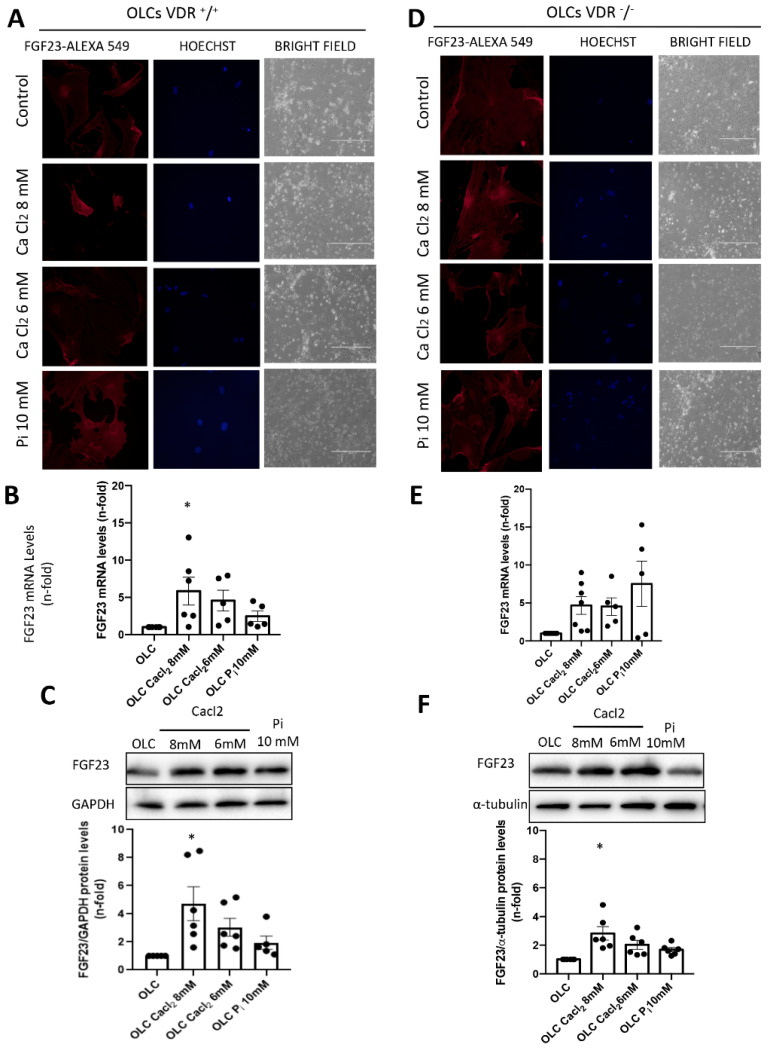
Study of the effects of calcium (CaCl_2_) and phosphorus (P_i_) on the gene and protein expression of FGF23. The OLCs were stimulated with different concentrations of calcium chloride (Cacl_2_) and inorganic phosphorus (Pi: NaH_2_PO_4_ + Na_2_HPO_4_) for 18 h. (**A**,**D**) FGF23 expression by fluorescence microscopy in CaCl_2_ and P_i_ stimulated VDR^+/+^ and VDR^−/−^ OLCs. (**B**,**E**) FGF23 gene expression in CaCl_2_ and P_i_ stimulated VDR^+/+^ and VDR^−/−^ OLCs by RT PCR. (**C**,**F**) The protein expression of FGF23 was also studied by Western blot. Data are expressed as the mean ± SEM of 4–6 independent experiments, * *p* < 0.05 vs. OLC without stimulus.

**Figure 3 nutrients-14-02576-f003:**
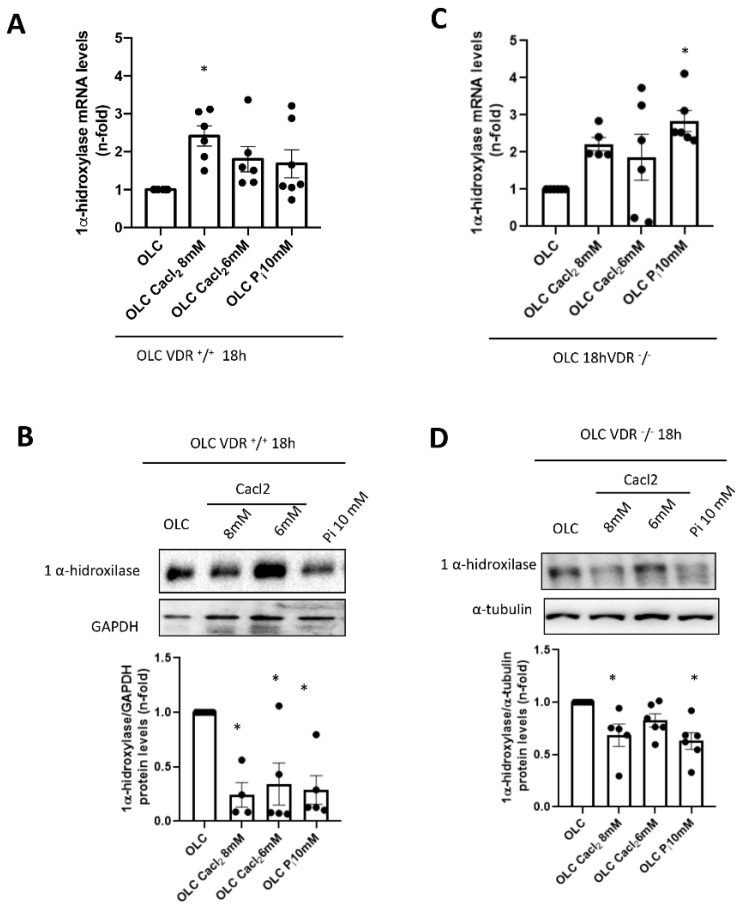
Study of the effects of calcium and phosphorus on the gene and protein expression of 1α-hydroxylase. The OLCs were stimulated with different concentrations of CaCl_2_ and inorganic phosphorus (P_i_) (NaH_2_PO_4_ + Na_2_HPO_4_) for 18 h. (**A**,**C**) The gene expression of 1α-hydroxylase was studied in VDR^+/+^ and VDR^−/−^ OLCs, respectively, by RT PCR. (**B**,**D**) FGF23 protein levels were also studied by Western blot at 18 h. Data are expressed as the mean ± SEM of 4–6 independent experiments, * *p* < 0.05 vs. OLCs without stimulus.

**Figure 4 nutrients-14-02576-f004:**
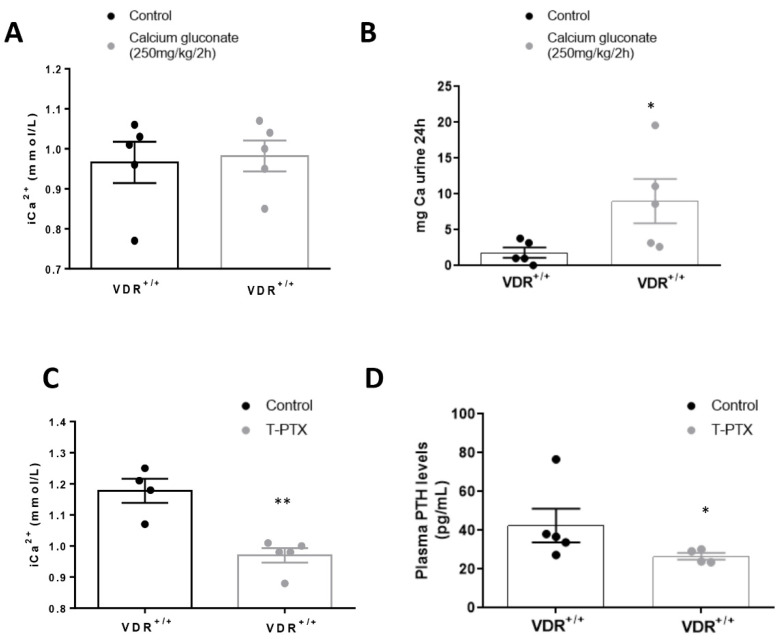
Establishment of a hypercalcemic model in VDR^+/+^ mice. To evaluate the role of Ca^2+^ in FGF23 synthesis, we established a mouse model with thyro-parathyroidectomy (T-PTX). VDR^+/+^ mice underwent a thyro-parathyroidectomy (T-PTX) induced by electro-cauterization under a dissecting microscope. After the surgery, mice received a daily subcutaneous replacement T4 treatment (L-thyroxine, 40 ng/g). Calcium levels were assessed in plasma samples (**A**) and 24 h urine samples (**B**). Blood samples 4 days after the surgery were taken to assess plasma Ca^2+^ (mmol/L) (**C**) and PTH levels (pg/mol) (**D**). Data are expressed as the mean ± SEM of 5–8 animals per group, * *p* < 0.05 vs. control values; ** *p* < 0.01 vs. control values.

**Figure 5 nutrients-14-02576-f005:**
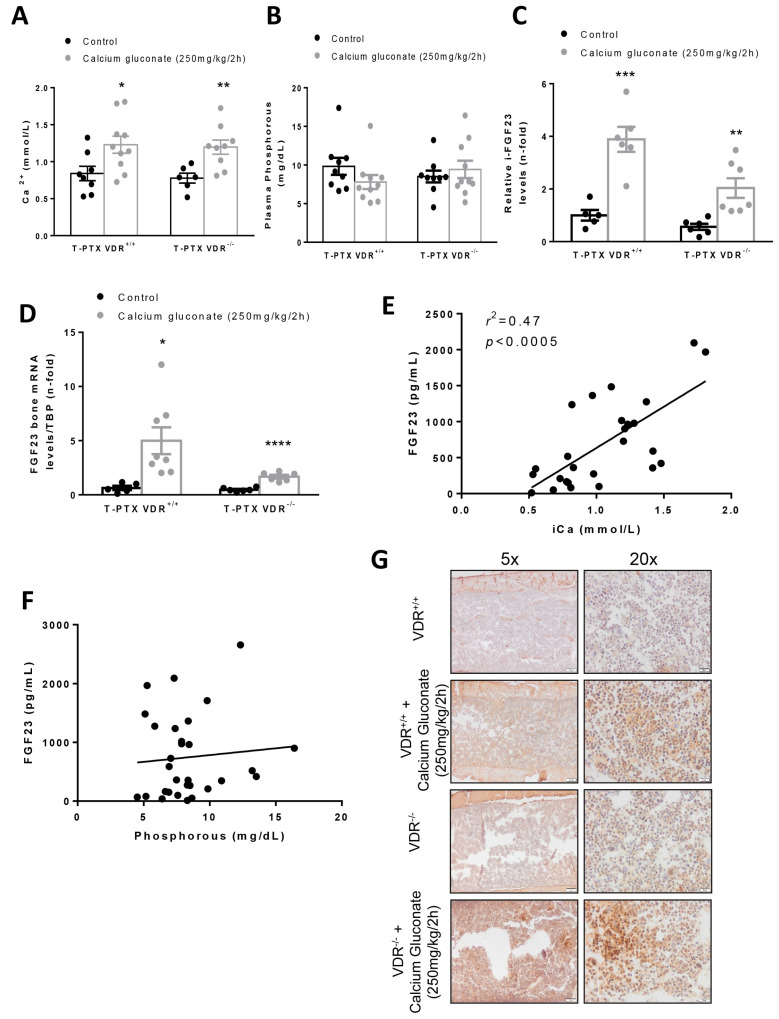
Experimental hypercalcemia in thyro-parathyroidectomized (T-PTX) VDR^+^/^+^ and VDR^−/−^ mice with constant levels of PTH modulates FGF23 levels. To evaluate the role of Ca^2+^ in the FGF23 synthesis without influence of PTH levels, we established a model of thyro-parathyroidectomy (T-PTX) VDR^+/+^ and VDR^−/−^ mice with subcutaneous implantation of osmotic minipumps filled with human PTH (1–34) (0.25 μL/h). After that, the animals were injected or not with calcium gluconate (250 mg/kg/2h; *n* = 10 mice per group) for 8 h. (**A**) Plasma Ca^2+^ levels in animals submitted to experimental hypercalcemia independent of PTH and VDR. (**B**) Total phosphorus plasma levels in T-PTX VDR^+/+^ and VDR^−/−^ with or without calcium gluconate injection. (**C**) Intact FGF23 plasma levels of T-PTX VDR^+/+^ and VDR^−/−^ mice injected with calcium gluconate. (**D**) Bone FGF23 gene expression assessed by RT-PCR. (**E**) Correlation of FGF23/Ca^2+^ in T-PTX VDR^+/+^ and VDR^−/−^ with and without calcium gluconate injection. (**F**) Correlation of FGF23/P in T-PTX VDR^+/+^ and VDR^−/−^ with and without calcium gluconate injection. (**G**) FGF23 protein levels in bone tissue samples of T-PTX VDR^+/+^ and VDR^−/−^ with and without calcium gluconate injection. * *p* < 0.05; ** *p* < 0.01; *** *p* < 0.005; **** *p* < 0.001 vs. T-PTX VDR^+/+^ or VDR^−/−^ without calcium gluconate injection. Data are expressed as the mean ± SEM of 5–8 animals per group.

**Figure 6 nutrients-14-02576-f006:**
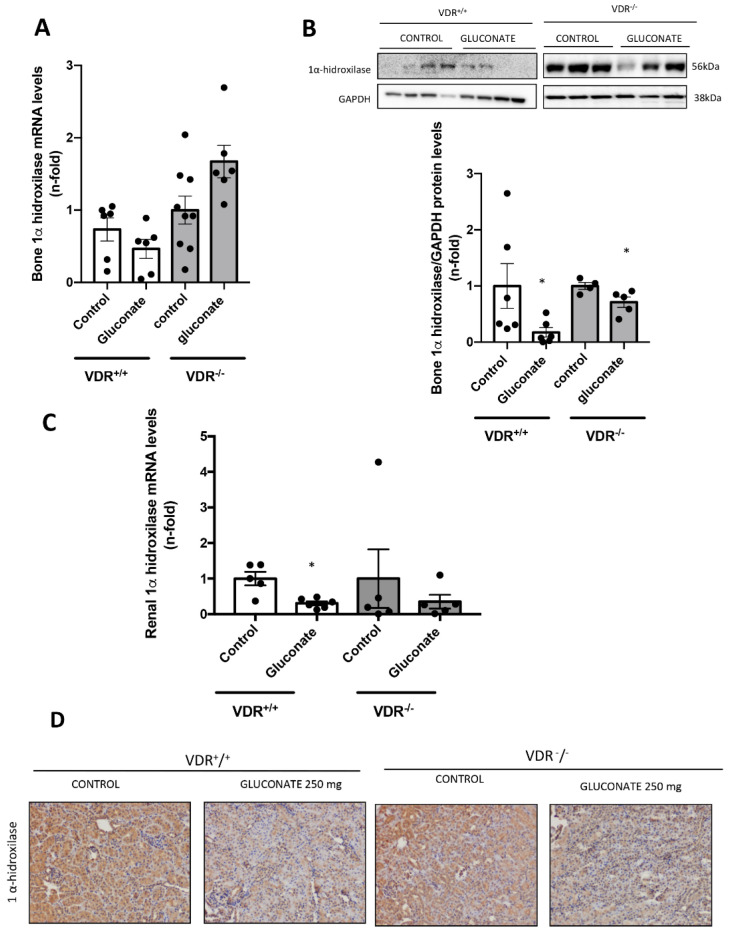
Study of the effects of calcium modulation on the gene and protein expression of 1α-hydroxylase in bone and renal tissue. PTX VDR^+/+^ or VDR^−/−^ mice were submitted to thyro-parathyroidectomy (T-PTX) by electro-cauterization. After this surgery, we infused mice with constant levels of PTH by osmotic minipumps. (**A**) Bone gene expression of 1α-hydroxylase in PTX VDR^+/+^ or VDR^−/−^ mice by RT PCR. (**B**) The bone protein expression of 1α-hydroxylase in PTX VDR^+/+^ or VDR^−/−^ mice by Western Blot. (**C**) Renal gene expression of 1α-hydroxylase in PTX VDR^+/+^ or VDR^−/−^ mice by RT PCR. (**D**) The 1α-hydroxylase protein expression in VDR^+/+^ or VDR^−/−^ mice by immunohistochemistry. Data are expressed as the mean ± SEM of 5–8 animals per group, * *p* < 0.05 vs. Control.

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
