# Peer review of "The Increase in FGF23 Induced by Calcium Is Partially Dependent on Vitamin D Signaling"

_nutrients, 2022, doi:10.3390/nu14132576_

Round 1
Reviewer 1 Report
Scientific contribution
In this study Rayego-Mateos et al explored the complex regulatory mechanism of FGF-23 and above all the regulation by calcium in relation to the presence or absence of VDRs. This is an important topic as understanding the regulation of the Klotho-FGF-23-Vitamin D axis would shed light on several mechanisms that are still poorly understood in the pathophysiology of hyperparathyroidism in patients with CKD.
MATERIAL AND METHODS
The study is very relevant as it evaluated the effect of FGF 23 in vivo and in vitro by also carrying out immunohistochemical analyzes at the level of the bone and kidney tissues of the mice examined. The performance of parathyroidectomy and the maintenance of constant PTH values through the implantation of a subcutaneous minipump is certainly very important, in this way it was possible to evaluate the effect of FGF 23 alone, excluding the PTH from the axis.
RESULTS
The results are very interesting. The results of section 3.1 should be explained in more detail especially in the text. In section 3.2 what explanation do you give regarding the non-modification of phosphate? It would also be interesting to have some data, if possible, on the expression of transporters such as NPT2a, NPT2c and TRPV5 at the renal tubular level.
Has Klotho's involvement been evaluated?
Discussion.
The discussion is clear, but it would be appropriate to clarify the results especially in vivo by giving your possible explanation of the FGF-23- Calcium-Vitamin D axis. Finally, the study is very interesting and needs only a minor revision with insights into the pathophysiology of CKD -MBD.
Author Response
Scientific contribution
Reviewer 1: In this study Rayego-Mateos et al explored the complex regulatory mechanism of FGF-23 and above all the regulation by calcium in relation to the presence or absence of VDRs. This is an important topic as understanding the regulation of the Klotho-FGF-23-Vitamin D axis would shed light on several mechanisms that are still poorly understood in the pathophysiology of hyperparathyroidism in patients with CKD.
Authors: Thank you so much for your assessment
MATERIAL AND METHODS
Reviewer 1: The study is very relevant as it evaluated the effect of FGF 23 in vivo and in vitro by also carrying out immunohistochemical analyzes at the level of the bone and kidney tissues of the mice examined. The performance of parathyroidectomy and the maintenance of constant PTH values through the implantation of a subcutaneous minipump is certainly very important, in this way it was possible to evaluate the effect of FGF 23 alone, excluding the PTH from the axis.
Authors: Thank you so much for your assessment
RESULTS
Reviewer 1: The results are very interesting. The results of section 3.1 should be explained in more detail especially in the text.
Authors: Thank you for the suggestion, we have elaborated a more detailed explanation of the results in the section 3.1
Reviewer 1: In section 3.2 what explanation do you give regarding the non-modification of phosphate?
Authors: The lack of effect of high P levels on the expression of 1α-hydroxylase in vitro is somewhat unexpected. Indeed, hyperphosphatemia will decrease active vitamin D synthesis in vivo. The decrease in active vitamin D production with high phosphate is likely offset by the ability of hyperphosphatemia to stimulate the secretion of FGF23, which will decrease the expression of 1α-hydroxylase. In our study, although we detected an increase in mRNA for 1α-hydroxylase in the incubation of OLC with high P, this was not translated into higher levels of protein. This result could imply that the effect of FGF23 in 1α-hydroxylase is post-transcriptional.
Reviewer 1: It would also be interesting to have some data, if possible, on the expression of transporters such as NPT2a, NPT2c and TRPV5 at the renal tubular level. Has Klotho's involvement been evaluated?
Authors: Your suggestion is very interesting. Indeed, we tried to evaluate also the renal levels of NPT2; TRPV5 and Klotho before submitting the manuscript, but we had methodological problems with the specificity of the antibodies against these transporters and factors, so the results there were inconclusive. In the time gave to respond to your comments (5 days) it is not possible to acquire new antibodies and set up the protocol to use it. In addition, we believe that these results will not necessary improve the message of the manuscript.
Discussion.
Reviewer 1: The discussion is clear, but it would be appropriate to clarify the results especially in vivo by giving your possible explanation of the FGF-23- Calcium-Vitamin D axis. Finally, the study is very interesting and needs only a minor revision with insights into the pathophysiology of CKD -MBD.
Authors: Thank you for your suggestion; we have implemented these modifications in the discussion.
Reviewer 2 Report
The authors in this article explored that “The increase in FGF23 induced by Calcium is partially dependent on vitamin D signaling”. It is an interesting article in the field. There are a few of concerns about the manuscript.
Major Concern:
1. The authors are suggested to perform dose-response effect of CaCl2 on FGF23 mRNA and protein expression in the OLC cultures. Then the optimal dose will be selected for the following experiments.
2. What is the purpose to use 10 mM Pi in the experiments? Is it a negative or positive control in the study design?
3. Were the MSCs cultured for 21 days without osteogenic stimuli? What is the culture media for MSCs?
Author Response
Comments and Suggestions for Authors
The authors in this article explored that “The increase in FGF23 induced by Calcium is partially dependent on vitamin D signaling”. It is an interesting article in the field. There are a few of concerns about the manuscript.
Major Concern:
Reviewer 2: The authors are suggested to perform dose-response effect of CaCl2 on FGF23 mRNA and protein expression in the OLC cultures. Then the optimal dose will be selected for the following experiments.
Authors: Thank you for your suggestion. In the manuscript we used 2 different doses of CaCl2 6mM and 8mM. These dose were based in a previous study develop in OLCs by David et al. in 2013. In this study the authors determined that both of these doses are the best option to increased FGF23 promoter levels after 12-18 hours, and finally increased FGF23 gene and protein expression levels (Endocrinology. 2013 Dec;154(12):4469-82). Therefore, we selected these doses and the best time point to perform the experiments (18 hours).
Reviewer 2:What is the purpose to use 10 mM Pi in the experiments? Is it a negative or positive control in the study design?
Authors: The use of inorganic phosphorus (Pi: NaH2PO4 + Na2HPO4) is a positive control that (theoretically) induces FGF23 expression in vitro.
Reviewer 2: Were the MSCs cultured for 21 days without osteogenic stimuli? What is the culture media for MSCs?
Authors: The cells that we used as control named in the manuscript as “MSCs” were cultured at the same time that (OLCs: differentiated MSCs) but without osteogenic stimuli during 21 days. We include a small explanation in the paragraph 2.2 entitled: In vitro studies.
The culture medium of MSCs without differentiation was αMEM supplemented with FBS (10%), ultraglutamine (1%), penicillin (100U/mL), and streptomycin (100 µg/ml).